# Self-evolving vision transformer for chest X-ray diagnosis through knowledge distillation

Sangjoon Park [1], Gwanghyun Kim[1], Yujin Oh [1], Joon Beom Seo[2], Sang Min Lee [2], Jin Hwan Kim[3], Sungjun Moon [4], Jae-Kwang Lim[5], Chang Min Park [6] & Jong Chul Ye [1,7 ✉]

Although deep learning-based computer-aided diagnosis systems have recently achieved expert-level performance, developing a robust model requires large, high-quality data with annotations that are expensive to obtain. This situation poses a conundrum that annually-collected chest x-rays cannot be utilized due to the absence of labels, especially in deprived areas. In this study, we present a framework named distillation for self-supervision and self-train learning (DISTL) inspired by the learning process of the radiologists, which can improve the performance of vision transformer simultaneously with self-supervision and self-training through knowledge distillation. In external validation from three hospitals for diagnosis of tuberculosis, pneumothorax, and COVID-19, DISTL offers gradually improved performance as the amount of unlabeled data increase, even better than the fully supervised model with the same amount of labeled data. We additionally show that the model obtained with DISTL is robust to various real-world nuisances, offering better applicability in clinical setting.

[1] Department of Bio and Brain Engineering, KAIST, Daejeon, Korea. [2] Asan Medical Center, University of Ulsan College of Medicine, Seoul, South Korea. [3] College of Medicine, Chungnam National Univerity, Daejeon, South Korea. [4] College of Medicine, Yeungnam University, Daegu, South Korea. [5] School of Medicine, Kyungpook National University, Daegu, South Korea. [6] College of Medicine, Seoul National University, Seoul, South Korea. [7] Kim Jaechul Graduate School of AI, KAIST, Daejeon, Korea. ✉email: jong.ye@kaist.ac.kr

With the early success of deep learning for medical imaging[1–3], the application of artificial intelligence (AI) for medical images has rapidly accelerated in recent years[4–6]. In particular, many deep learning-based computer-aided diagnosis (CAD) software have been introduced into routine practice[7–10] for various imaging modalities including chest X-ray (CXR). These deep learning-based AI models have demonstrated the potential to dramatically reduce the workload of clinicians in a variety of contexts if used as an assistant, leveraging their power to handle a large corpus of data in parallel. The advantage can be maximized in resource-limited settings such as in underdeveloped countries, where various diseases like tuberculosis prevail while the number of experts to provide accurate diagnosis is scanty.

Currently, most of the existing AI tools are based on the convolutional neural network (CNN) models built with supervised learning. However, collecting large and well-curated data with the ground truth annotation is rather difficult in the underprivileged areas where the amount of available data itself is abundant. In particular, although the size of data increases in number every year in these areas, the lack of ground truth annotation hinders the use of increasing data to improve the performance of AI model. Given the limitation in label availability, an important line of machine learning research is to obtain a robust model relying less on manually annotated labels.

Several classes of approaches have been developed to address this problem, by learning the underlying patterns of data without or with only a small amount of pre-existing labels. In self-supervised learning, the large unlabeled data corpus itself provides the supervisory signal which enables the model to learn task-agnostic visual representation and to adapt to the downstream tasks with a small number of labeled data. In self-training[11], which is a representative semi-supervised learning method, a learner (teacher) obtained with supervised learning via small labeled data keeps on labeling large unlabeled data, generating pseudo-labels, which can be used for retraining a new model (student) with enlarged data corpus. This teacher-student learning paradigm is often called knowledge distillation. Since this configuration is suitable for the framework with a siamese design where one model learns from the prediction of the other model instead of labels, some lines of semi- and self-supervised works have utilized knowledge distillation and suggested the possibility that the model performance can be similar or even better than the fully supervised one[11–13].

Specifically, in self-training with noisy student method[11] (Fig. 1a), the key idea is to match the predictions of a more corrupted student to the pseudo-label obtained with an uncorrupted teacher. Specifically, in this method, the teacher is first trained with supervised learning on the small number of labeled data to generate the pseudo-labels for the separated set of large unlabeled data corpus. During the training of the student with both labeled and pseudo-labeled data, strong noise is applied both for the input and model architecture of the student to improve the robustness to nuisances like adversarial or natural perturbations. These processes are iteratively done a few times by treating the trained student as a new teacher to generate new pseudo-labels for unlabeled data. In distillation with no label (DINO)[12] (Fig. 1b), two networks with the same Vision Transformer (ViT) models[14], one defined as a student and one as a teacher, take inputs from two sets of views from the same image: two large patches containing the global idea of the image (global crops), and the multiple small patches that offer a local representation of the image (local crops). Then, all crops are passed to the student model while only the global crops are passed to the teacher so that the two networks come to understand the semantic meaning with self-attention that the local and the global crops represent the same subject, albeit seemingly disparate. More detailed explanation of the existing approaches is provided in Supplementary material and Supplementary Fig. 1.

Although self-training with noisy student and DINO are seemingly different, they share fundamental similarities of knowledge distillation that the teacher outputs more accurate result with less distorted or more informative input, and the student learns to match the teacher's prediction using more distorted or less informative input. Focusing on this, here we introduce a self-evolving framework, dubbed distillation for self-supervised and self-train learning (DISTL), that can gradually improve the performance by the generation of pseudo-labels that reconcile the distinct strengths of self-supervised learning and self-training under knowledge distillation.

Figure 1c illustrates the proposed DISTL framework. The two identical models, teacher and student, are utilized. However, different from the existing methods, the student is jointly trained with self-supervised learning for semantic features and self-training for pseudo-labels with the knowledge distilled from the teacher. These two components play different roles in the learning process. Specifically, self-supervised learning empowers the model to learn task-agnostic semantic information about the given image, like overall structural features consisting the CXR image (Supplementary Fig. S2). On the other hand, the self-training enables the model to directly learn the task-specific information such as the diagnosis of tuberculosis, by encouraging the student to match its noised prediction obtained from a given CXR to the clean pseudo-label of the teacher. In addition, inspired by the learning process of human non-expert reader, the additional correction with the initial small labeled data is done per the pre-defined steps to prevent the student from being biased by the imperfect estimation of the teacher. More detailed descriptions for the DISTL method and ablation study to verify the architecture are provided in Supplementary Material and Supplementary Fig. S3.

With the proposed DISTL method, we have shown that the AI model performance can be gradually improved with increasing unlabeled data by maximally utilizing the common ground of knowledge distillation from self-supervision and self-training. Of note, it even outperforms the supervised model trained with the same amount of labeled data. Furthermore, the model obtained with the DISTL method has shown to be substantially robust to the real-world data corruptions as well as offers a more straightforward localization of lesions with the model's attention. We argue that the distillation of knowledge through self-supervised learning and self-training, even without knowledge of the lesion, leads to a high correlation of attention with the lesion, which may also be the reason for the superior diagnostic performance.

## Results

**Practical simulation for increasing unlabeled data over time.** Figure 2 illustrates the experimental scenario of clinical application of the proposed DISTL framework. With the small number of labeled data, the initial model is first obtained, and the student model is trained in large unlabeled data with this teacher. During this process, the teacher is slowly co-distilled from the updated student. Then, the updated models are used as the starting points of the next-generation models with increasing time $T$, similar to the self-training with noisy student[11]. We evaluated the proposed framework in three CXR tasks including the diagnosis of tuberculosis by using only a small corpus of labeled data for supervision and gradually increasing the amount of unlabeled data simulating the real-world data accumulation over time.

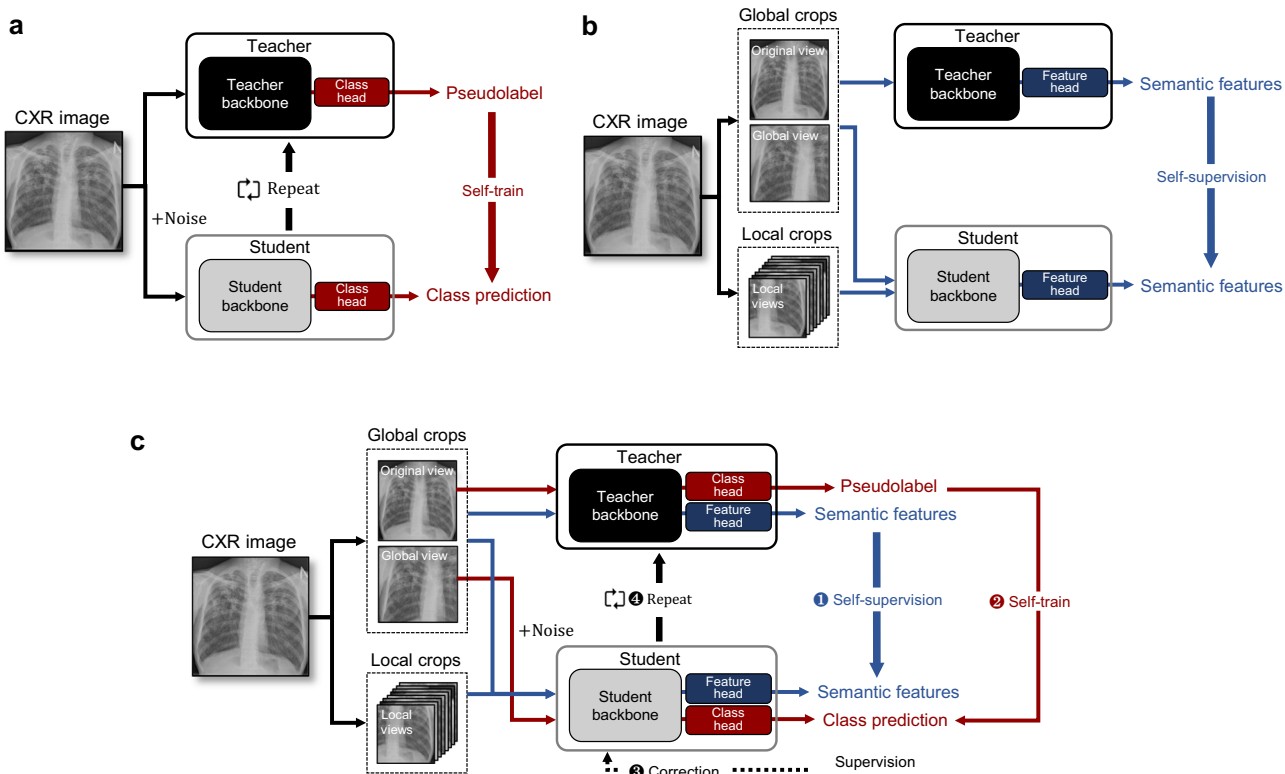

**Fig. 1 Comparison between the existing methods and the proposed framework for the self-evolving AI model. a** Self-training with noisy student method. **b** Self-supervised learning with distillation with no label (DINO) method. **c** The proposed distillation for self-supervised and self-train learning (DISTL) method is mainly composed of the two components, for self-supervision and self-training.

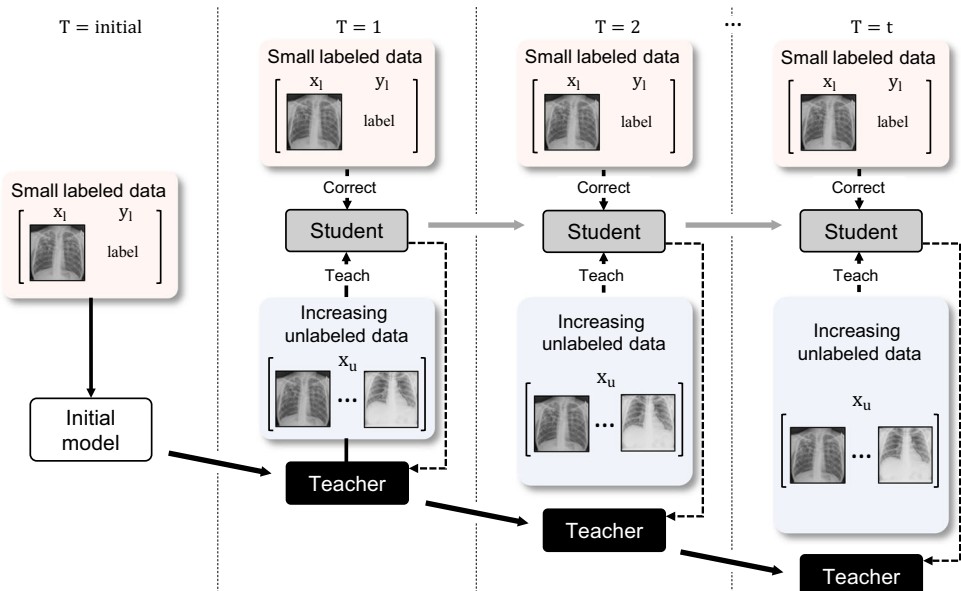

**Fig. 2 Simulation of clinical application for increasing data over time.** The initial model is trained with small labeled data. Then, using this initial model as teacher and small initial data for correction, the student is trained with the DISTL method under an increasing amount of data over time $T$.

In particular, to confirm whether our AI model can gradually self-evolve in the data-abundant but the label-insufficient situation, we set our main task as the diagnosis of tuberculosis, as it is highly demanded in clinics after World Health Organization has identified the use of AI-based CAD for CXR screening of tuberculosis as a potential solution in resource-limited settings[15]. We collected the normal and tuberculosis CXRs from both the publicly available open-source and the institutional datasets for the model development and internal validation (Supplementary Table 1 and Details of datasets for diagnosis section). After collection, a total of 35,985 CXRs were further divided into 3598 labeled (10% of total data) and 32,387 unlabeled subsets (90% of total data). Next, assuming the situation in the clinic that the number of unlabeled cases

**Table 1 Details of data partitioning used for the experiments of three chest X-ray (CXR) tasks.**

| Class | Training and internal validation | | | | External validation | | | |
|---|---|---|---|---|---|---|---|---|
| | $T$ = initial | $T$ = 1 | $T$ = 2 | $T$ = 3 | | | | |
| | Labeled | Unlabeled | | | Total | CNUH | YNU | KNUH |
| *Tuberculosis diagnosis* | | | | | | | | |
| Normal | 3033 | 9013 | 18,047 | 27,059 | 1100 | 400 | 300 | 400 |
| Tuberculosis | 565 | 1782 | 3543 | 5328 | 328 | 28 | 100 | 200 |
| Total | 3598 | 10,795 | 21,590 | 32,387 | 1428 | 428 | 400 | 600 |
| *Pneumothorax diagnosis* | | | | | | | | |
| Normal | 984 | 2920 | 5816 | 8726 | 1100 | 400 | 300 | 400 |
| Pneumothorax | 224 | 707 | 1438 | 2155 | 120 | 30 | 40 | 50 |
| Total | 1208 | 3627 | 7254 | 10,881 | 1220 | 430 | 340 | 450 |
| *COVID-19 diagnosis* | | | | | | | | |
| Normal | 3015 | 9017 | 17,999 | 27,077 | 1100 | 400 | 300 | 400 |
| COVID-19 | 503 | 1538 | 3111 | 4590 | 659 | 81 | 286 | 292 |
| Total | 3518 | 10,555 | 21,110 | 31,667 | 1759 | 481 | 586 | 692 |

*T* time, *CNUH* Chungnam national university hospital, *YNU* Yeungnam University Hospital, *KNUH* Kyungpook National University Hospital; *COVID-19* coronavirus disease 2019.

increases as time goes, the unlabeled subset was further divided into three. Then, using these three folds, we increased the total amount of available unlabeled data to be 30%, 60%, 90% of total data, supposing the time $T = 1, 2, 3$ goes. During this process, the subset of labeled data remains fixed to the initial 3598 CXRs (10% of total data). The performances of the proposed self-evolving AI model at each time $T$ were evaluated in the external validation data collected and labeled by board-certified radiologists in three different hospitals (Chonnam National University Hospital [CNUH], Yeungnam University Hospital [YNU], and Kyungpook National University Hospital [KNUH]), to validate the generalization capability for different devices and image acquisition settings (Table 1, upper).

For pneumothorax diagnosis, we used the SIIM-ACR pneumothorax data[16] for the model development and internal validation. As it contains the CXRs and the segmentation mask for either pneumothorax or normal cases, we adopted it to be the pneumothorax diagnosis task, as a binary classification problem. Similar to the tuberculosis diagnosis task, we partitioned this data into a labeled and unlabeled subsets, and the unlabeled subset was further divided into three to simulate the gradually accumulating data with time. For external validation of the trained model, we also collected the CXRs of pneumothorax patients in the three hospitals (CNUH, YNU, KNUH), which were labeled by board-certified radiologists (Table 1, middle).

For COVID-19 diagnosis, we utilized two publicly available COVID-19 datasets[17,18] for the model development and internal validation by gradually increasing the amount of unlabeled data with increasing time similar to other tasks, while the CXRs of polymerase chain reaction (PCR) positive COVID-19 cases were deliberately collected for the external validation in the three hospitals (CNUH, YNU, KNUH) (Table 1, lower).

**Our tuberculosis diagnosis model can self-evolve with increasing unlabeled data.** We first investigated whether the performance of tuberculosis diagnosis can gradually be improved with the proposed DISTL framework given the increasing numbers of unlabeled data. As shown in Fig. 3a, in the external validation, the performance of the model trained with the proposed framework improved as the number of unlabeled data increased, from an AUC of 0.948 to 0.974. Of note, the improved performance was even better than the supervised model trained with the same amount of data with labels, which improved to the AUC of 0.958 at $T = 2$ but decreased to 0.950 at $T = 3$, showing

the sign of overfitting. In detail, the final model showed the AUCs of 0.974, 0.965, 0.985, 0.980, sensitivities of 92.7%, 92.9%, 93.0%, 95.0%, specificities of 92.0%, 90.3%, 96.0%, 93.5%, accuracies of 92.2%, 90.4%, 95.3%, 94.0% in the pooled test set and three institutions (Supplementary Table 2), which confirmed the excellent generalization capability in clinical situation with difference devices and settings.

Not confined to the metric itself, we observed an interesting finding that the model attention of the ViT model gets refined with increasing time $T$ (Fig. 3b). As the AI model evolves with increasing time $T$, the self-attention of AI gets refined to better localize the target lesion as well as semantic structures within the given CXR image. Notably, the gradual improvement of performance was more prominent for the ViT model equipped with self-attention than the standard CNN-based models without self-attention (Fig. 4a). The ViT model showed a linear increase as well as the best performance among the models, although other CNN-based models also showed performance improvement with the proposed framework under increasing unlabeled data. In addition, the ViT model showed no sign of overfitting which was observed in some CNN-based models at later $T$.

We further evaluated whether the existing self-supervised and semi-supervised learning methods, which can also be utilized for the plenty of unlabeled data with increasing $T$, can improve the performance of the AI model gradually similar to our DISTL framework (Fig. 4b). With the same experimental settings, the existing methods presented the significant degradation of performance at $T = 1$ where the number of unlabeled data is relatively small, while the performances slightly improve with more data with increasing $T$. Even with this increase in performance, none of the existing self-supervised and semi-supervised methods showed prominent performance improvement compared with the initial model, while the model built with the DISTL framework showed stably improving performance with increasing unlabeled data.

**Robustness to unseen classes and label corruption.** In a real-world setting, the data of unseen classes not included in training data may be included when collecting the unlabeled cases, or incorrectly labeled data may be added by the mistake of a practitioner. Therefore, we performed experiments to verify the robustness of the proposed framework given these situations. First, we collected data on four other classes (nodule, effusion, interstitial lung disease, bacterial infection) that are commonly

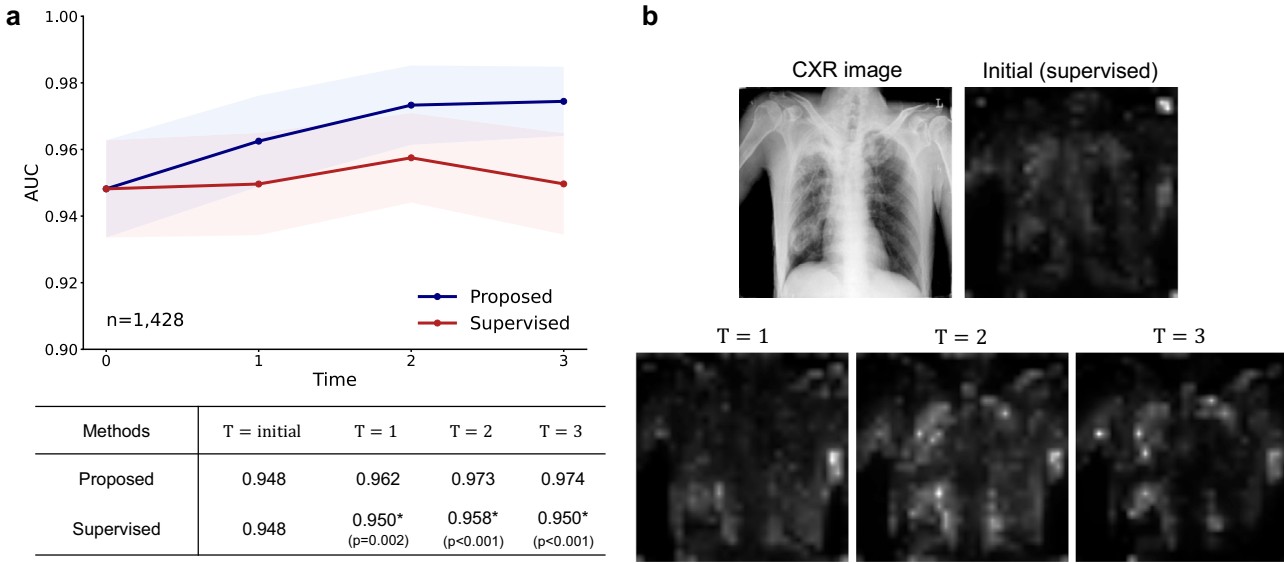

**Fig. 3 Tuberculosis diagnosis results with gradually increasing unlabeled data. a** Gradually evolving performance with the proposed framework under an increasing amount of data. Compared with the supervised model using the same amount of data with labels, the model trained with the proposed framework without labels showed even better performances. **b** Gradual attention change of the evolving model for a tuberculosis case. For the exemplified tuberculosis case, the attention of the Vision Transformer (ViT) model gets gradually refined to better catch the target lesions and semantic structures as the model evolves with increasing time $T$. Data are presented with calculated area under the receiver operating characteristics curves (AUCs) in the study population (center lines) ±95% confidence intervals (CIs) calculated with the DeLong's method (shaded areas). The AUCs of the proposed method and the supervised learning method were compared at each time point $T$ with the DeLong test to evaluate statistical significance, except for the $T = initial$ where the two methods start from the same baseline. * denotes statistically significant ($p < 0.050$) superiority of the proposed framework. All statistical tests were two-sided. CXR, chest X-ray.

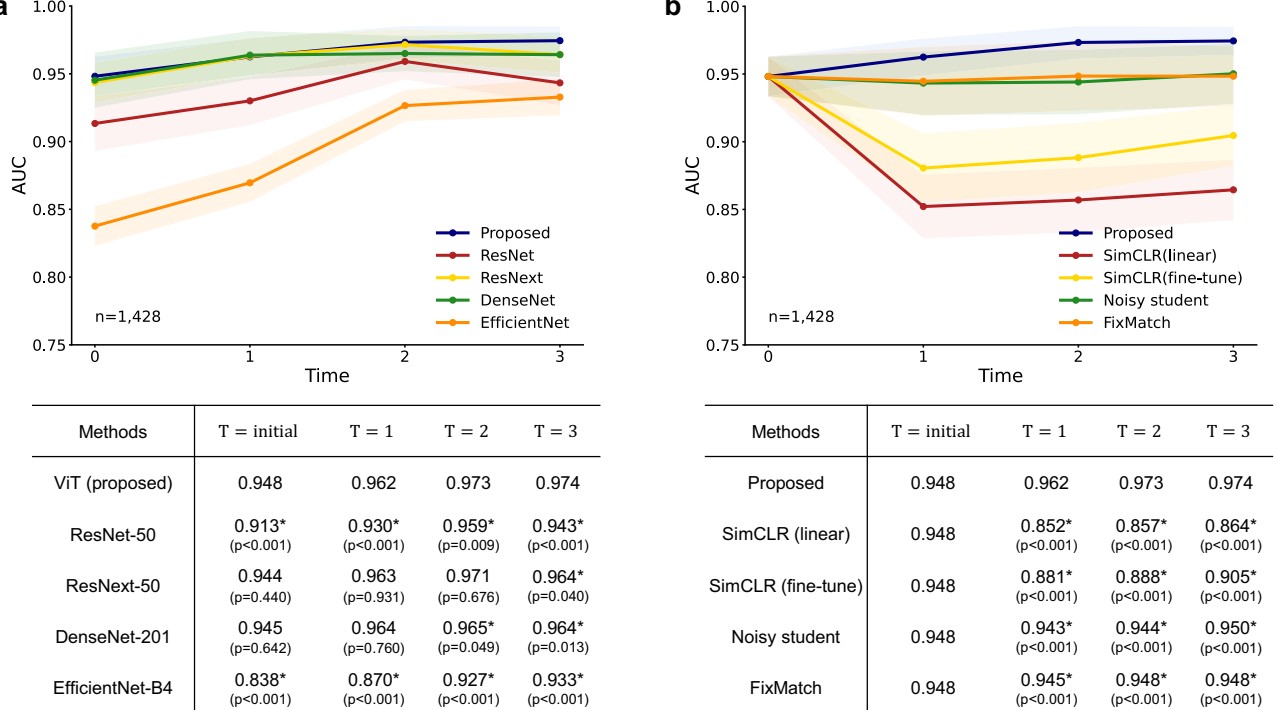

**Fig. 4 Tuberculosis diagnosis results for comparison with the other models and methods. a** Compared to other convolutional neural networks (CNN)-based models, the Vision Transformer (ViT) model showed a linear increase as well as the best performance among the models. **b** Unlike the model trained with the proposed framework, none of the existing self-supervised and semi-supervised methods showed a prominent improvement in performance with increasing time $T$. Data are presented with calculated area under the receiver operating characteristics curves (AUCs) in the study population (center lines) ±95% confidence intervals calculated with the DeLong's method (shaded areas). The AUCs of the proposed method were compared with the other models or methods at each time point $T$ with the DeLong test to evaluate statistical significance, except for the $T = initial$ where all methods of comparison start from the same baseline in **b**. * denotes statistically significant ($p < 0.050$) superiority of the proposed framework. All statistical tests were two-sided.

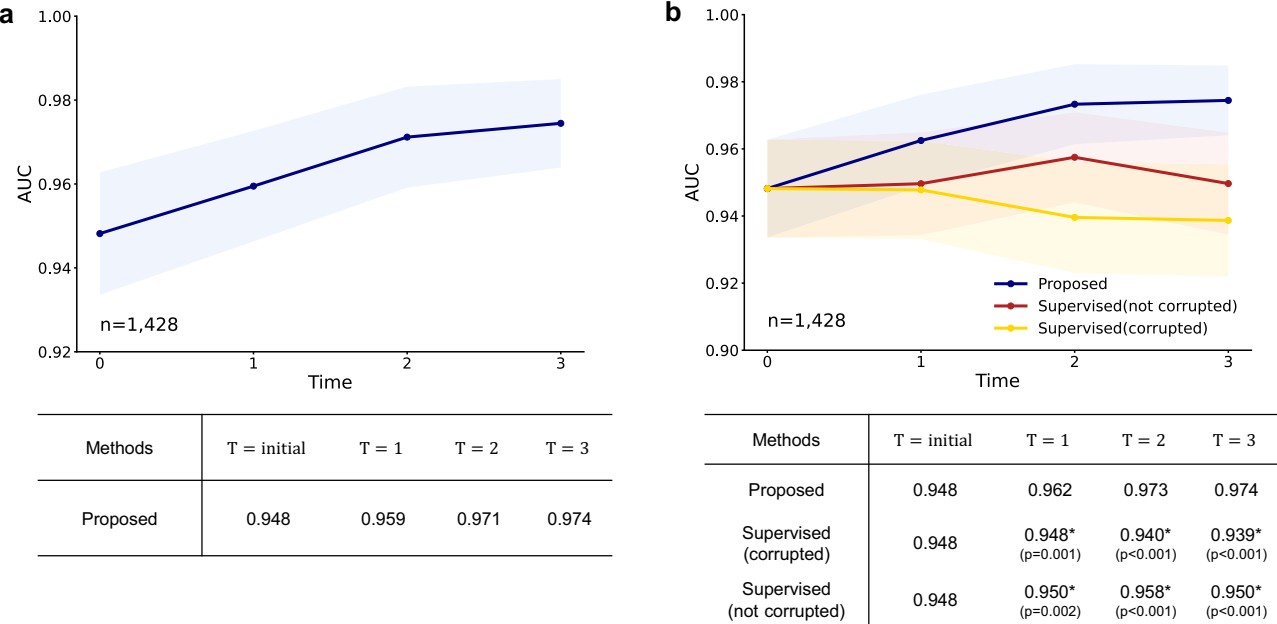

**Fig. 5 Tuberculosis diagnosis results with real-world data collection and label corruption. a** Even after adding unseen class data that are commonly encountered in clinics, the performance was stably improved with increasing time *T*, even though these other class data were not included for the training of the initial model. **b** In the simulation for label corruption, the model trained with the proposed framework was not compromised, while the model trained with supervised learning using corrupted labels showed significant deterioration in performance. Data are presented with calculated area under the receiver operating characteristics curves (AUCs) in the study population (center lines) ±95% confidence intervals calculated with the DeLong's method (shaded areas). The AUCs of the proposed method and the supervised learning method with and without corruptions were compared at each time point *T* with the DeLong test to evaluate statistical significance, except for the *T = initial* where all methods start from the same baseline. * denotes statistically significant ($p < 0.050$) superiority of the proposed framework. All statistical tests were two-sided.

encountered in clinics from a hospital (Asan Medical Center [AMC]). These other class data were added in the same manner when increasing the number of unlabeled data over time (Supplementary Table 3). Notably, the performance was stably improved the same as in the experiments without adding these other class data (Fig. 5a), suggesting the robustness of the proposed framework to assure that the AI model is not confused by these unseen classes to the initial model trained only with normal and tuberculosis data. Secondly, we randomly made the label wrong with a probability of 5% for the supervised learning and evaluated whether the performance decreases (Supplementary Table 4). The model trained with supervised learning using the corrupted label showed significant deterioration in performance, while that with the proposed framework was not altered as it does not depend on the label for increased data (Fig. 5b). Taken together, these results suggest the impressive reliability of the proposed framework which is required in real clinical applications.

**Comparison of lesion localization performances.** Under the hypothesis that ViT's direct attention can provide better localization than CNN's indirect attention via the Gradient-weighted Class Activation Mapping (GradCAM)[19], we quantified the localization performance with model attention. A total of 30 CXRs in the external validation data for tuberculosis diagnosis were selected, and manually annotated by the clinician. The predictions from model attention were generated by applying the threshold values after normalization to best localize the target lesions (0.1 for ViT and 0.6 for CNN models). As the ViT model has multiple heads to be visualized, the best performing head was selected for evaluation. The dice similarity coefficients were calculated to assess the consistency between the predictions and labels.

Without any supervision during the training, the direct visualization of ViT attention offered better localization of the target lesion than the indirect attention visualization of CNN-based models using GradCAM, providing a mean dice similarity coefficient of 0.622 (standard deviation [STD] of 0.168) compared with that of 0.373 (STD of 0.259) for a CNN-based model. Of note, the indirect attention using GradCAM either attends to the unimportant location (upper figure) or fails to localize the multiple lesions (lower figure) (Fig. 6).

**Verifying applicability of the proposed framework in other tasks.** We further analyze whether the gradual performance improvement with the proposed framework can also be observed in the CXR tasks other than tuberculosis diagnosis. First, for pneumothorax diagnosis, similar to the observation in the tuberculosis diagnosis task, the model trained with the proposed framework improved gradually over increasing time *T* (Fig. 7a). Notably, the performance of the model with the proposed framework was lower than the supervised one when available unlabeled data are relatively small (*T* = 1) but it ultimately outperformed the supervised model with the increased numbers of unlabeled data (*T* = 3). Similarly, for COVID-19 diagnosis, the proposed framework provided a stable performance improvement over time, whereas the model trained with the same amount of labeled data showed a substantial performance drop at later *T* in the external validation (Fig. 7b), suggesting that overfitting to training data degraded the generalization performance of the supervised model.

**Validation in a cohort of bacteriological laboratory test confirmed tuberculosis cases.** As the diagnostic gold standard of tuberculosis is bacteriological laboratory test, we further validated

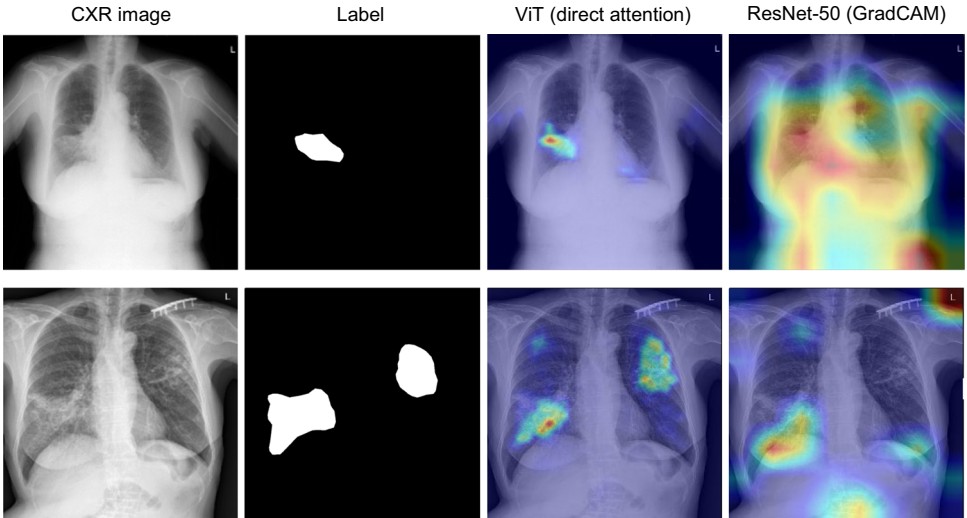

**Fig. 6 Exemplified tuberculosis lesion localization with attention by Vision Transformer (ViT) and convolutional neural network (CNN)-based models.**
The direct visualization of ViT attention offers better localization of the target lesion than indirect attention visualization of CNN-based models using Gradient-weighted Class Activation Mapping (GradCAM).

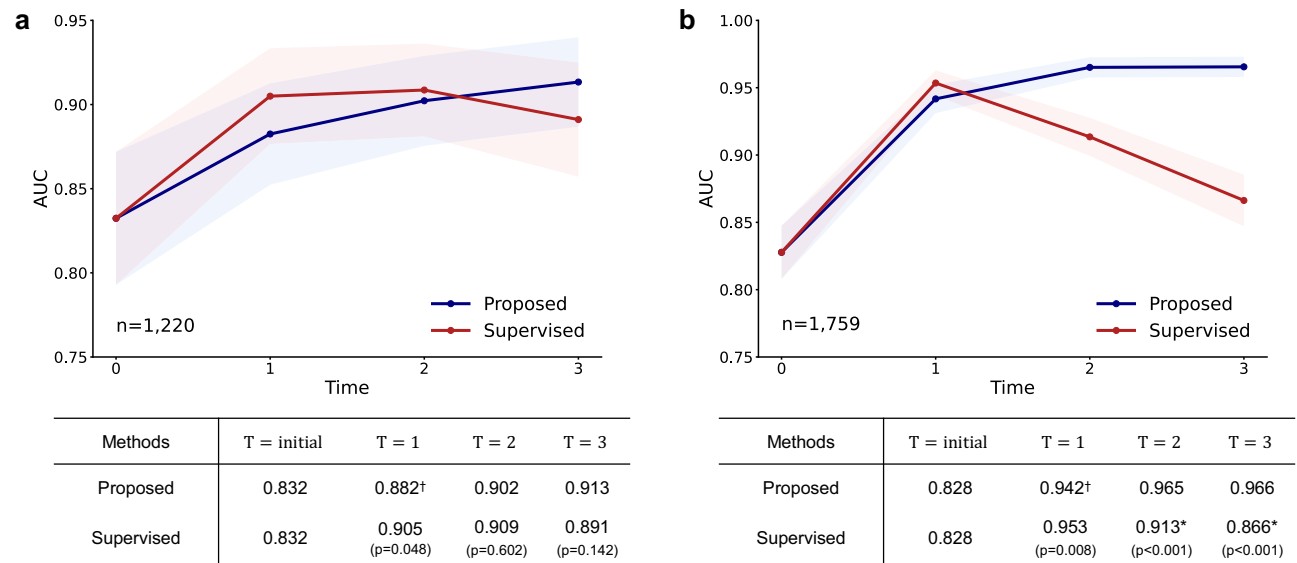

| Methods | T = initial | T = 1 | T = 2 | T = 3 |
|---|---|---|---|---|
| Proposed | 0.832 | 0.882† | 0.902 | 0.913 |
| Supervised | 0.832 | 0.905 (p=0.048) | 0.909 (p=0.602) | 0.891 (p=0.142) |

| Methods | T = initial | T = 1 | T = 2 | T = 3 |
|---|---|---|---|---|
| Proposed | 0.828 | 0.942† | 0.965 | 0.966 |
| Supervised | 0.828 | 0.953 (p=0.008) | 0.913* (p<0.001) | 0.866* (p<0.001) |

**Fig. 7 Other CXR diagnosis results including pneumothorax and coronavirus disease 2019 (COVID-19) diagnoses. a** When applied for pneumothorax diagnosis, the model trained with the proposed framework improved gradually over increasing time $T$, while the supervised model showed the sign of overfitting at later $T$. **b** Likewise, the model trained with the proposed framework outperformed the supervised model for the COVID-19 diagnosis task, which was more prominent under the increasing amount of unlabeled data. Data are presented with calculated area under the receiver operating characteristics curves (AUCs) in the study population (center lines) ±95% confidence intervals calculated with the DeLong's method (shaded areas). The AUCs of the proposed method and the supervised learning method were compared at each time point $T$ with the DeLong test to evaluate statistical significance, except for the $T = initial$ where the two methods start from the same baseline. † denotes statistically inferior performance, while * denotes statistically significant superiority of the proposed framework ($p < 0.050$). All statistical tests were two-sided.

the developed model in the external data from another corelab (Seoul National University Hospital [SNUH]) containing confirmed tuberculosis cases with the reference standard. To this end, CXR images of 727 normal and 535 bacteriological laboratory test confirmed tuberculosis cases were collected. In this cohort, the results were remarkably consistent. Given the increasing number of unlabeled data, the model obtained with DISTL frameworks provided gradually improving performance without any sign of overfitting even at later $T$, which was superior to those of a fully supervised model trained with the same amount of labeled data (Fig. 8). In detail, the final model with the DISTL framework

showed a diagnostic performance with an AUC of 0.952, a sensitivity of 87.7%, specificity of 89.1%, and accuracy of 88.5% (Supplementary Table 5) demonstrating the robust generalization performance in cohort with the reference standard.

## Discussion
Given the impressive results of early studies that AI can keep up or even surpass the performance of the experienced clinician in various applications for medical imaging[1,2], we have confronted the era of flooding AI models for medical imaging. However,

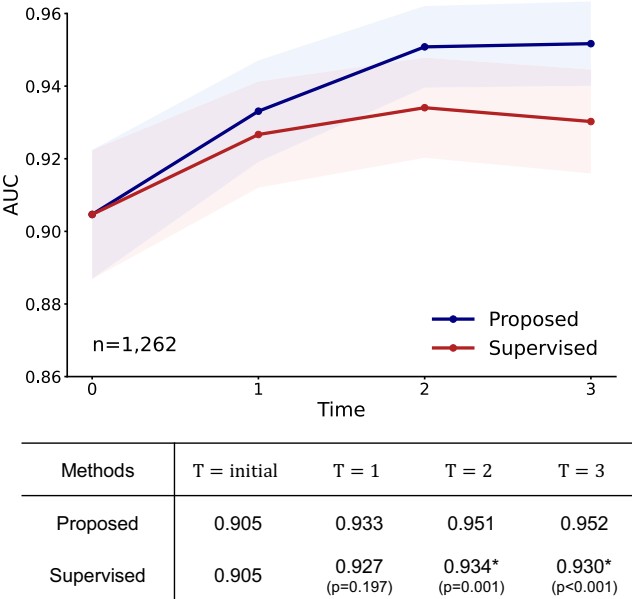

| Methods | T = initial | T = 1 | T = 2 | T = 3 |
|---|---|---|---|---|
| Proposed | 0.905 | 0.933 | 0.951 | 0.952 |
| Supervised | 0.905 | 0.927 (p=0.197) | 0.934* (p=0.001) | 0.930* (p<0.001) |

**Fig. 8 Tuberculosis diagnosis results for validation in a cohort of bacteriological laboratory-confirmed tuberculosis cases.** In a cohort from another corelab consisting of 727 normal cases and 535 confirmed tuberculosis cases, the model obtained with the DISTL framework offered performances superior to those of the fully supervised model trained with the same amount of labeled data. Data are presented with calculated area under the receiver operating characteristics curves (AUCs) in the study population (center lines) ±95% confidence intervals (CIs) calculated with the DeLong's method (shaded areas). The AUCs of the proposed method and the supervised learning method were compared at each time point T with the DeLong test to evaluate statistical significance, except for the T = initial where the two methods start from the same baseline. * denotes statistically significant (p < 0.050) superiority of the proposed framework. All statistical tests were two-sided.

these models share a common drawback that they highly depend on the quantity and quality of labels as well as the data. If the labeled corpus does not contain sufficient data points to represent the entire distribution, the resulting model can be biased and the generalization performance can unpardonably deteriorate. In the field of medical imaging, a large number of raw data is being accumulated each year, but it is difficult to utilize this large data corpus with supervised learning approaches due to the absence of labels. Therefore, several methods based on unsupervised learning[20,21], self-supervised learning[22], and semi-supervised learning[23] have been proposed to cope with this problem, but their performances are still sub-optimal.

To cope with this problem, the proposed DISTL framework stands based on two key components: self-supervised learning and noisy self-training through teacher-student knowledge distillation. The first component, in our method, is similar to that proposed in DINO[12], which encourages the model to learn the task-agnostic semantic information of the image by the local-global view correspondence. In our preliminary experiment, the model built only with this self-supervision attends well to the image layout, and particularly, object boundaries as shown in Supplementary Fig. S2. Secondly, the self-training component enables the model to directly learn the task-specific features, the diagnosis of tuberculosis, similar to the noisy student self-training[11]. Under the continuity and clustering assumption[24], learning with a soft pseudo-label along with student-side noise increases not only the performance but also the robustness to adversarial samples.

Interestingly, we have found an analogy between the proposed DISTL framework and the training process of radiologists during their junior years (Fig. 9). When a junior radiologist learns to read CXR, a common practice is to first read CXR and affirm it with computed tomography image of the same patient which usually offers a more accurate diagnosis. This procedure is analogous to the learning process of the student in our framework in which the model learns to match the prediction from noisy augmented or cropped image containing less information to that from the clean original image containing more information by the teacher which offers a more accurate diagnosis. In addition, it is also a common practice that the junior radiologist learns referring to the senior radiologist's reading, which is similar to the teacher-student distillation used in our framework. Finally, during the learning process, the junior radiologists occasionally refer to the textbook containing small but typical cases, which prevents from being biased from recently seen atypical cases. In our framework, the correction step with the small number of initial labeled data plays a similar role. As a result, the DISTL framework, unlike the existing self-supervised and semi-supervised learning approaches, offered gradually evolving performance simply by increasing the amount of unlabeled data, with the substantial robustness to the corruptions from data of unseen classes or wrong labels. In addition, we found in the experiments extending the application of our framework to the pneumothorax and COVID-19 diagnosis that it provides the benefit generally applicable to a variety of tasks.

Practically, our method holds great potential for the screening of diseases like tuberculosis, especially when applied in underprivileged areas. In the simulation of the application of the proposed framework, it shows a negatively predicted portion of 72.5%, a negative predictive value (NPV) of 0.977, and a false-positive rate (FPR) of 0.080 (Supplementary Table 6). That is to say, 72.5% of the screened population are indeed negative with a probability of 97.7%, and therefore can be safely excluded from the further reading by the clinician, having only 8.0% of cases as false positive. Consequently, this will substantially reduce the workload, while not significantly increasing the false alarm. Moreover, the AI model can self-evolve with the proposed DISTL method in the increasing unlabeled data corpus without any further supervision from a human expert. This is another important merit to be used in underprivileged areas, where plenty of data are available due to the high prevalence of diseases but the experts to label data are scanty.

This study has several limitations. First, the details concerning the patient demographics and CXR characteristics were not available in open-source data used for the training and internal validation. Second, although we simulated the robustness to unseen class data assuming real-world data collection in the experiment, it was not possible to consider all the other minor classes that can be encountered in real-world data accumulation. Third, we utilized a total of 35,985 CXRs to demonstrate the benefit of the proposed framework by using after dividing them into the small labeled and large unlabeled subsets, but the number may be insufficient to draw a firm conclusion. Further studies are warranted to verify the proposed framework in a data corpus large enough to represent the general distribution. Nevertheless, with the data-abundant but label-insufficient condition being common for medical imaging, we believe that it may offer great applicability to a broad field of medical imaging.

## Methods
**Ethics committee approval.** This study was approved by the institutional review boards (IRBs) of Asan Medical Center, Chungnam National University Hospital, Yeungnam University Hospital, Kyungpook National University Hospital, Seoul

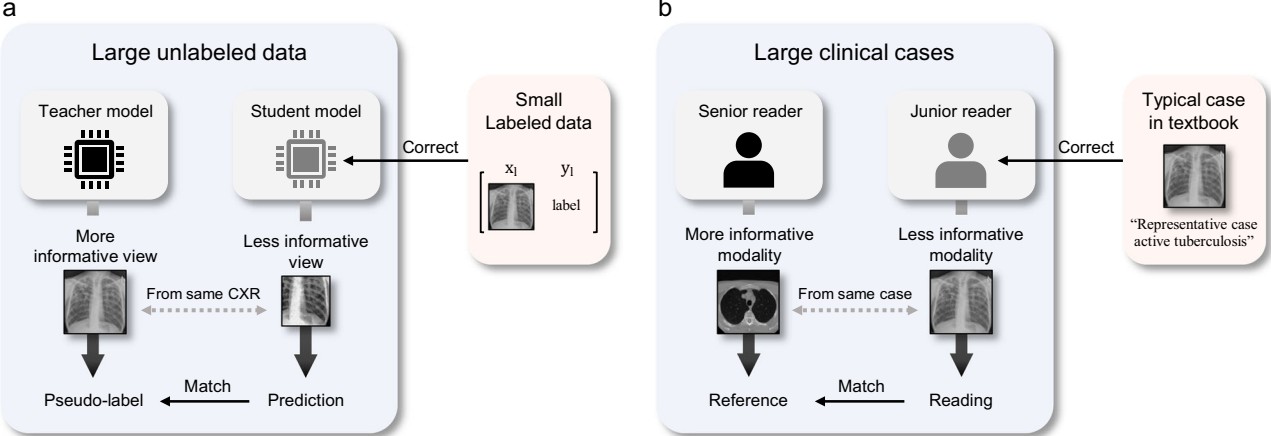

**Fig. 9 The analogy between the artificial intelligence (AI) model training with the proposed DISTL method and human reader training. a** The learning process of AI model trained with the proposed DISTL method. **b** The learning process of a radiologist. CXR, chest X-ray.

National University Hospital, and the requirement for informed consent was waived due to the retrospective study design.

**Details of datasets for pre-training**. In this work, the pre-trained model was used for all models, to offer the feature extracting capacity with prior knowledge of CXR. To pre-train the model to learn task-relevant CXR feature in a large corpus of CXRs, we used the CheXpert dataset[25] containing 10 common CXR classes: no finding, cardiomegaly, lung opacity, consolidation, edema, pneumonia, atelectasis, pneumothorax, pleural effusion, and support device. Among the 10 classes, five classes including lung opacity, consolidation, edema, pneumonia, and pleural effusion, considered to be related to the manifestation of infectious disease, were selected as task-relevant CXR features. Consequently, the model was first trained to classify these five classes with the CheXpert data. With a total of 223,648 CXRs from 65,240 subjects, 29,453 posterior-anterior (PA) and 161,759 anterior-posterior (AP) view CXRs were used after excluding the 32,436 lateral view CXRs. Thanks to the huge number of cases, the model was able to be a robust extractor for the task-relevant CXR features, without depending upon the variation in patients and the setting for image acquisition. As suggested in the ablation study of pre-training (see Supplementary Fig. S3), this pre-training step has brought us a substantial increase in performance and is one of the key components of our model.

**Details of datasets for diagnosis**. First, for the tuberculosis diagnosis, we used both public and institutional datasets for the model development and internal validation. Specifically, data deliberately collected from a hospital (Asan Medical Center [AMC]) as well as publicly available data (National Institutes of Health [NIH][26], Valencian Region Medical ImageBank [BIMCV][17], CheXpert[25], India[27], Montgomery, Shenzen[28], Belarus[29], PAthology Detection in Chest radiograph [PADChest][30], Tuberculosis X-ray 11K [TBX 11K][31]) were integrated (Supplementary Table 1), to have a total of 35,985 CXRs containing 5893 tuberculosis and 30,092 normal cases. For pneumothorax diagnosis, the SIIM-ACR pneumothorax data[16] were utilized for the model development and internal validation. The original SIIM-ACR pneumothorax data contains 12,089 CXR images with or without corresponding masks for the 2379 pneumothorax or 9710 normal cases. Therefore, we adapted the problem into a binary classification for the diagnosis of pneumothorax by defining CXR as a pneumothorax positive case if a segmentation mask contains a positive value and as a negative case if not. Finally, for COVID-19 diagnosis, two publicly available open-sourced datasets were used for the model development and internal validation[17,18]. As these two datasets contain a small number of normal CXRs, we also utilized the normal CXRs in the tuberculosis diagnosis task as the normal cases, finally yielding a total of 35,185 CXRs consisting of 5093 COVID-19 and 30,092 normal cases.

For all the three CXR diagnosis tasks, data deliberately collected from three hospitals (Chonnam National University Hospital [CNUH], Yeungnam University Hospital [YNU], and Kyungpook National University Hospital [KNUH], labeled by board-certified radiologists, were used to externally validate the generalization capability for different devices and image acquisition settings. In detail, 328 tuberculosis and 1100 normal CXRs for the tuberculosis diagnosis task, 120 pneumothorax and 1100 normal CXRs for the pneumothorax diagnosis task, and 120 COVID-19 and 1100 normal CXRs are evaluated for the external validation of each task.

**Details of Implementation**. The CXR images underwent preprocessing including histogram equalization, Gaussian blurring, and normalization, and resized to $256 \times 256$. As the backbone part $f$ of the network, we used the ViT small model (12

layers and six heads) with the patch size of $8 \times 8$, and the CNN-based models (ResNet-50, ResNext-50, DenseNet-201, and EfficientNet-B4) were used for comparison. For the classification and projection heads, the three-layered multi-layer perceptron was utilized. For pre-training, an Adam optimizer was used with a learning rate of 0.0001. The model was pre-trained for 5 epochs with a step decay scheduler, with a batch size of 16. Weak data augmentation including random flipping, rotation, and translation were performed to increase the data variability during the pre-training[32]. As the loss functions, binary cross-entropy (BCE) losses were used for each class label. For supervised training of the initial model and the iterative training of the models with the DISTL method, AdamW optimizer[33] were used along with cosine decay scheduler with a maximum learning rate of 0.00005. The model was trained for 5 epochs with one epoch for the warm-up step. The correction step is performed per 500 updates. Similar to pre-training, a batch size of 16 was used and random data augmentation was performed during the training. Since the differences between the image classes in CXR are less discernible than those in natural image, the cropped area should be larger to capture the distinctive information between the CXRs. Therefore, we used a larger range of crop window sizes for both global and local crops with 75–100% and 20–60% of the entire image, respectively, compared with those of 40–100% and 5–40% for the DINO method. The cross-entropy loss was used as the loss function for both the classification and the self-supervising losses. All experiments including preprocessing, model development, and evaluation were performed using Python 3.8.5, Pytorch 1.8.0, Numpy 1.22.2, Pillow 9.0.1, Opencv-python 4.5.5.62, timm 0.5.4, scikit-learn 1.0.2 libraries and CUDA 11.1 on NVIDIA Quadro 6000, GeForce RTX 3090, and RTX 2080 Ti.

**Details of evaluation**. For evaluation of the overall model performances, the three independent test sets were pooled and then used to evaluate the overall performance of the model, while the model performances in three individual test sets were reported separately. The area under the receiver operating characteristics curve (AUC) was used as the primary evaluation metric, and the sensitivity, specificity, and accuracy were also calculated to meet the pre-defined sensitivity value ≥80% by adjusting the thresholds, if possible. To statistically compare the proposed method with others, the DeLong test was performed to estimate 95% confidence intervals (CIs) and $p$-values. Statistically significant differences were defined as $p < 0.050$. All statistical tests were two-sided.

Direct visualization of the model attention is another merit of the ViT model. Similar to the approach introduced in a self-supervised learning approach for ViT[12], we used the attention weights of multi-head in the last layer of the Transformer encoder to visualize attention. For comparison, the model attention was visualized indirectly with the Grad-CAM[19] that generates the model attention with the linear combination determined by the gradients of the output with regard to the last layer feature map, for the CNN-based models.

**Reporting summary**. Further information on research design is available in the Nature Research Reporting Summary linked to this article.

## Data availability

Part of CXRs is compiled from publicly available open-source data repositories. The CheXpert repository is available at https://stanfordmlgroup.github.io/competitions/chexpert/, The BIMCV repository is available at https://github.com/BIMCV-CSUSP/BIMCV-COVID-19. The India tuberculosis repository can be found at https://www.

kaggle.com/raddar/chest-xrays-tuberculosis-from-india. Montgomery and Shenzen data can be requested via the contact on the follwing webpage https://openi.nlm.nih.gov. Belarus tuberculosis repository is available at https://github.com/frapa/tbcnn/tree/master/belarush. The PADChest repository is available at https://github.com/auriml/Rx-thorax-automatic-captioning. The TBX 11K repository can be accessed at https://www.kaggle.com/usmanshams/tbx-11. NIH normal data can be found at http://cloud.google.com/healthcare-api/docs/resources/public-datasets/nih-chest and NIH tuberculosis data can be available at https://tbportals.niaid.nih.gov/downloaddata after getting permission from TB portal. The SIIM-ACR Pneumohtorax Segmentation dataset is available at https://www.kaggle.com/c/siim-acr-pneumothorax-segmentation. Brixia COVID-19 data repository can be found at https://brixia.github.io/. Other part of institutional data, which were used with institutional permission through IRB approval for this study, are not publicly available due to the patient privacy obligation. Interested users can request the access to these data for research, by contacting the corresponding author J.C.Y. (jong.ye@kaist.ac.kr). Any access to de-identified institutional data requires IRB approval at the requesting institution along with the signed agreement on data transfer and usage. Replies to initial request will be made within 10 working days and follow-up based on the answers will be made within institutional review cycles. Use of data is limited to research purposes and redistribution of data is not allowed. Source data are provided with this paper.

## Code availability

The Pytorch codes for the DISTL used in this study is available at the following Github repository at https://doi.org/10.5281/zenodo.662325[34].

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

## Acknowledgements

We are grateful to Yisak Kim (Seoul National University Hospital) for his help in collecting validation data. This research was supported by the National Research Foundation (NRF) of Korea under Grant NRF-2020R1A2B5B03001980, the Korea Medical Device Development Fund grant funded by the Korea government (the Ministry of Science and ICT, the Ministry of Trade, Industry and Energy, the Ministry of Health & Welfare, the Ministry of Food and Drug Safety) (Project Number: 1711137899, KMDF_PR_20200901_0015), Institute of Information & communications Technology Planning & Evaluation (IITP) grant funded by the Korea government (MSIT) (No. 2019-0-00075, Artificial Intelligence Graduate School Program (KAIST)), Institute of Information & communications Technology Planning & Evaluation (IITP) grant funded by the Korea government (MSIT) (No. 2021-0-02068, Artificial Intelligence Innovation Hub), the MSIT (Ministry of Science and ICT), Korea, under the ITRC (Information Technology Research Center) support program (IITP-2022-2020-0-01461) supervised by the IITP (Institute for Information & communications Technology Planning & Evaluation) and the KAIST Key Research Institute (Interdisciplinary Research Group) Project.

## Author contributions

S.P. performed all experiments, wrote the extended code, and prepared the manuscript. G.K. and Y.O. contributed in data preprocessing. J.B.S., S.M.L., J.H.K., S.M., and J.K.L. collected and labeled data. C.M.P. advised the project in conception. J.C.Y. supervised the project in conception and discussion, and prepared the manuscript.

## Competing interests

The authors declare no competing interests.
