## [Peer Review File · Nature Communications]

Reviewers' Comments:

Reviewer #1:

Remarks to the Author:

This paper presents a semi-supervised deep learning method for chest x-ray image analysis. The authors proposed to use a recent vision transformer deep learning architecture coupled with knowledge distillation framework for better chest x-ray diagnosis. The proposed method was applied to address multiple problems including the diagnosis of tuberculosis, pneumothorax and COVID-19. The results show that the proposed method had better performances than other fully supervised (ResNet, DenseNet and Efficient-Net) or self-supervised approaches (SimCLR).

This paper addresses an important and challenging problem in medical imaging. Self-supervised learning (and semi-supervised learning) enables us to learn meaningful image features without relying on the availability of labelled data. However, it was difficult to clearly understand the contributions of the paper due to the poor quality of writing. Many terms mentioned in Introduction were not clearly explained and motivation of using knowledge distillation framework was not also well articulated.

While I appreciate the authors for conducting comprehensive experiments with multiple diseases and data sources from 3 different hospitals, I am not fully convinced whether this would be enough for a publication at Nature Communication. I think the paper still requires a lot more work to improve writing and articulate the contributions of the paper.

1. what is knowledge distillation, self-supervised learning and self-training? These terms are not either defined or not clearly explained.
2. What are the teacher and noisy student ?
3. The proposed method appears to be same as the method called distillation with no label (DINO). Did authors come up with any modifications or changes to improve chest x-ray image analysis?

Reviewer #2:

Remarks to the Author:

In the article "AI can evolve without labels: self-evolving vision transformer for chest X-ray diagnosis through knowledge distillation", the authors presented a new DL framework using knowledge distillation through self-supervised learning and self-training. The authors demotivated the performance and the application of the framework by building three models for TB, pneumothorax and Covid using training sets that are mostly composed of unlabelled data. External validations were conducted and showed the proposed model outperformed supervised learning models for all three diseases. Overall, this paper is very well written - a clear description of the model and methods. The results have important applications in medical imaging AI.

Specific comments:

Abstract

In my opinion, I was expecting to see some mention of the model construction and external validation in the abstract and the results. I was directly given the conclusion that the new framework has good application in covid, TB and pneumothorax. So I would ask to revise the abstract as above within the limit.

Results

Radiological assessment is not a definitive measure in terms of diagnosing TB, it seems the reference standard for TB in the external validation dataset is radiologist reading rather than bacteriological lab results. This is a limitation.

The reference standards were mentioned for the external dataset of covid and TB on page 7, but no description of how pneumothorax was determined. I only found it in "Details of datasets for diagnosis" section. It would be good to include the reference standards all in the same place.

Figure-4, 5, and 8: Can you plot 95% CI as error bars?

Code availability: the Github link does not work.

NCOMMS-22-04319A

Response to the Reviewers

General Comments

We thank the editor and reviewers for the constructive reviews. Your comments have helped us to significantly improve our work. To address the comments, we have done the following works and made major changes accordingly:

1. The detailed explanations of related works, terms, and motivation of our method were provided in the Introduction section. Figures and contexts are revised accordingly.
2. For better understanding, figures comparing the existing methods (*self-training with noisy student* and *distillation with no label [DINO]*) were added.
3. Additional external validation was performed in another institution, with the cohort consisting of bacteriological laboratory test confirmed tuberculosis cases. The results were remarkably consistent with our previous results.
4. Abstract section has been revised to include the context of model construction and external validation results.
5. We added a new figure describing the analogy between the learning process of the proposed method and that of the radiologist-in-training, to better explain the motivation of our method.
6. Error bars showing the 95% confidence interval were added to all graphs.
7. We revised the overall structure and order of text for better readability.

Reply to the Reviewer 1

This paper presents a semi-supervised deep learning method for chest x-ray image analysis. The authors proposed to use a recent vision transformer deep learning architecture coupled with knowledge distillation framework for better chest x-ray diagnosis. The proposed method was applied to address multiple problems including the diagnosis of tuberculosis, pneumothorax and COVID-19. The results show that the proposed method had better performances than other fully supervised (ResNet, DenseNet and Efficient-Net) or self-supervised approaches (SimCLR).

This paper addresses an important and challenging problem in medical imaging. Self-supervised learning (and semi-supervised learning) enables us to learn meaningful image features without relying on the availability of labelled data. However, it was difficult to clearly understand the contributions of the paper due to the poor quality of writing. Many terms mentioned in Introduction were not clearly explained and motivation of using knowledge distillation framework was not also well articulated.

While I appreciate the authors for conducting comprehensive experiments with multiple diseases and data sources from 3 different hospitals, I am not fully convinced whether this would be enough for a publication at Nature Communication. I think the paper still requires a lot more work to improve writing and articulate the contributions of the paper.

→ Thanks for the constructive comments. We agree with the reviewer’s comment that the background knowledge, terms, and our motivation for the proposed method should be explained in more detail, considering the broad readership of “*Nature Communications*”. Therefore, we have provided a more detailed explanation of related works, terms, and the motivation of our method in the Introduction section of text, Terminology section of Supplementary material, and Fig. S1. The overall context is also revised for better readability. Besides, we added a figure explaining the differences between the existing self-supervised, self-training methods and our method in Fig. 1. Another inspiring figure explaining the analogy between the learning process of the proposed method and the human reader is provided in Fig. 10. We believe that this revision has significantly improved the clarity of our paper. We appreciate the reviewer’s comment.

R1C1. what is knowledge distillation, self-supervised learning and self-training? These terms are not either defined or not clearly explained.

→ Thanks for the constructive comments. We added a detailed explanation of backgrounds and terms in the Introduction section, Supplementary material, Fig. S1, and reorganized the text.

Specifically, to address the difficulties in obtaining the manually annotated labels, two important lines of work are unsupervised learning and semi-supervised learning, which leverages plenty of unlabeled data and relies less on the label. Below Fig. R1 illustrates the detailed description and relationship for the concepts used in this work.

Fig. R1. Illustration of learning methods. The colored circles denote labeled data, while the empty circles denote unlabeled data. (a) Supervised learning with fully labeled data and (b) with the small number of labeled data. (c) Self-supervised with the unlabeled data. Similar data points are clustered without any supervision from label. (d) Semi-supervised learning with the small labeled and large unlabeled data. Unlabeled data are utilized referring to the labeled data points, as painted in faint colors that stand for the pseudo-labels. Given the small number of labels, self- or semi-supervised learning provides more accurate decision boundary than supervised learning.

More specifically, unsupervised learning is a learning approach that learns underlying patterns of data without any pre-existing label. Self-supervised learning is a type of unsupervised learning approach, in which the unlabeled data itself provides the supervisory signal which enables the model to learn underlying patterns of data. Usually, it is formulated from a large data corpus by training the model to learn several pretext tasks, which usually includes matching the differently augmented versions of the same image¹, predicting mask area², matching the original image from small cropped lesion³ and so on. Through the pretext learning, the model can learn to discriminate between similar and dissimilar data, and thereby cluster them without any supervision from label (Fig. R1c). After the self-supervised learning, the model can easily adapt to the downstream tasks with a small number of labeled data, as it has already learned task-agnostic visual representation from the pretext tasks.

Semi-supervised learning, as the name suggests, is a learning paradigm located somewhere between supervised and unsupervised learning. It is generally used when a small amount of labeled data and a large amount of unlabeled data exist. In semi-supervised learning, a large number of unlabeled data is utilized by referring to the small amount of labeled data. For example, a large number of unlabeled data points can be used to provide a more accurate decision boundary for all data for the classification task (Fig. R1d). Self-training is one of the representative semi-supervised learning methods. In this method, a learner (teacher) obtained with supervised learning in small labeled data keeps on labeling large unlabeled data, called *pseudo-labels*. The pseudo-labels, in turn, constitutes an enlarged data corpus, so that one can use the data for retraining the model (student)⁴. In earlier works, it has been shown to be a promising approach to leverage unlabeled data to improve model performances⁵⁻⁷.

Knowledge distillation is a learning paradigm of transferring the knowledge from the teacher model to the student model, as mentioned above in the explanation of self-training. It was originally developed for model compression, where the aim is to efficiently build the simple student by distilling the knowledge of the complex teacher, often required for practical implementation of AI model in devices with limited computational resources⁸. However, since this configuration can be utilized in the framework with a siamese design where one model learns from the prediction of the other model instead of labels, some lines of semi- and self-supervised learning works utilized knowledge distillation as mentioned above in the application for self-training. Of note, several recent studies have suggested the possibility that the model can obtain a performance similar to or better than the fully supervised model through semi- or self-supervised learning methods based on the knowledge distillation framework^{3,6}.

R1C2. What are the teacher and noisy student?

→ "*Self-training with noisy student*" method, which is one of the key components of our framework, is a consistency-based learning method that successfully utilized the knowledge distillation for semi-supervised learning⁶. The key idea of the method is to match the predictions of a more corrupted student to the pseudo-label obtained with an uncorrupted teacher. In this method, two models referred to as "teacher" and "noisy student (referred to as student)" are built, either with the same architecture or a more complex one for the student than the teacher. Then, the teacher is first trained with supervised learning on the small number of labeled data. Second, the teacher model generates the pseudo-labels for the separated set of large unlabeled data corpus, and the student model is trained using both labeled and unlabeled data, by leveraging the real label and pseudo-label, respectively. During the training of the student, as the name "*noisy student*" suggests, strong *noise* is applied both for the input and model architecture of the student (see Fig. R2). Specifically, for the input *noise*, random augmentations⁹ including geometric transform, and contrast adjustment are used, while the random dropout¹⁰ and stochastic model depth¹¹ are used as the architectural *noise*. Applying the strong *noise* to the student during training and enforcing it to match the pseudo-label of the clean teacher makes the student model to be robust to nuisances like adversarial or natural perturbations, which leads to the enhanced generalization performances in the real-world setting. Moreover, these processes are iteratively done a few times by treating the trained student as a new teacher to generate new pseudo-labels for unlabeled data and training a new student using these. By iterating such procedure, the performance of the final

model can gradually improve. A detailed illustration of the "*Self-training with noisy student*" is provided in below Fig. R2.

Fig. R2. Illustration of *Self-training with noisy student* method.

We added this context in the manuscript and added a figure providing the detailed method of self-training with noisy student in Fig. 1a.

R1C3. The proposed method appears to be same as the method called distillation with no label (DINO). Did authors come up with any modifications or changes to improve chest x-ray image analysis?

→ Thanks for the constructive comment. *DINO* is a noticeable method that applied knowledge distillation for self-supervised learning, which has demonstrated impressive performance exploiting the knowledge distillation through the view correspondence. Fig. R3 below illustrates the *DINO* method. In detail, two networks with the same architecture, one defined as a student and one as a teacher, will take inputs from two sets of views from the same image: two large patches containing the global idea of the image are cropped (global crops), and the multiple small patches that offer a local representation of the image are cropped (local crops). Then, all crops are passed to the student model while only the global crops are passed to the teacher. During the training, the student model is trained with the less informative local crops to match the prediction of the teacher obtained with the more informative global crops. Consequently, the two networks come to understand that the local and the global crops represent the same subject, albeit seemingly disparate.

Fig. R3. Illustration of Self-supervised learning with *distillation with no label (DINO)* method.

Our *distillation for self-supervised and self-train learning (DISTL)* framework started from the intuition that "*self-training with noisy student*" and "*DINO*" are seemingly different, but share fundamental similarities of knowledge distillation that the teacher outputs more accurate results with less distorted or more informative input, and the student learns to match the teacher's prediction using more distorted or less informative input.

In the *DISTL* framework, the two identical models, teacher and student are utilized similar to *DINO*. However, different from *DINO*, the student is jointly trained with self-supervised learning and self-training with the knowledge distilled from the teacher. In fact, the generation and use of pseudo-labels for the self-training is one of the key differences between the two methods.

Fig. R4. Illustration of the *distillation for self-supervised and self-train learning (DISTL)* method.

In detail, Fig. R4 illustrates our *DISTL* framework. Although *DINO* is one of the key components of our framework (component (1) in above Fig. R4), our *DISTL* method is different from *DINO* in several aspects. First, the proposed *DISTL* also depends on the task-specific supervisory signal from *self-training with noisy student* method, by matching the class prediction of the student to pseudo-label generated by teacher (component (2) in above Fig. R4) which enables the model to learn the task-specific information like the diagnosis of tuberculosis, while the *DINO* method only learns task-agnostic semantic information about the given image, like overall structural features consisting the CXR image. Second, unlike *DINO* where the training is performed in an end-to-end fashion without iterative learning, our *DISTL* method is performed iteratively (component (4) in Fig. R4) given the increasing unlabeled data corpus, making the trained student and teacher models as starting points for the next-generation model to provide gradual performance improvements as in below Fig. R5.

Fig. R5. Simulation of clinical application for increasing data over time. The initial model is trained with small labeled data. Then, with this model as the teacher and small initial data for correction, the student is trained with the *DISTL* method under an increasing amount of data over time T .

Third, inspired by the learning process of inexperienced radiologists who occasionally revise their knowledge through the small number of “representative cases” in the textbook to prevent from being biased to atypical cases confronted, the “correction step (component (3) in Fig. R4)” with small labeled data is also adopted in our method to prevent student model from being biased by the imperfect estimation of the teacher. This component is not included in the *DINO* as well as in the *self-training with noisy student*. Since this component was inspired by the training process of a radiologist, the figure describing an analogy between the learning processes of the *proposed* method and radiologist-in-training was added as in below Fig. R6.

Fig. R6. The analogy between (a) the artificial intelligence (AI) model training with the proposed *DISTL* method and (b) human reader training.

Some other detailed modifications were adopted for the implementation tailored for CXR image analysis. First, the pre-processing including histogram equalization, Gaussian blurring, and normalization were performed for all input CXR images. Second, all models in this work started from the pre-trained weights from CheXpert data¹² to provide the feature extracting capacity with prior knowledge of CXR. Finally, we used larger range of crop window sizes for both global and local crops with 75-100% and 20-60% of the entire image, respectively, compared with those of 40-100% and 5-40% for the original *DINO* method. Since the differences between the image classes in CXR is less discernible than those in natural image, the cropped area should be larger to capture the distinctive information between the CXRs.

We added the comparison between the *DINO* and the proposed *DISTL* in Fig. 1b-c, and figure describing the analogy in Fig. 10. Other detailed implementations for chest X-ray images are also added in the Methods section of the text.

Reply to the Reviewer 2

In the article “AI can evolve without labels: self-evolving vision transformer for chest X-ray diagnosis through knowledge distillation”, the authors presented a new DL framework using knowledge distillation through self-supervised learning and self-training. The authors demontated the performance and the application of the framework by building three models for TB, pneumothorax and Covid using training sets that are mostly composed of unlabeled data. External validations were conducted and showed the proposed model outperformed supervised learning models for all three diseases. Overall, this paper is very well written - a clear description of the model and methods. The results have important applications in medical imaging AI.

→ We appreciate the reviewer’s understanding and encouraging comments. We have made our point-by-point responses as below.

Specific comments:

R2C1. (Abstract) In my opinion, I was expecting to see some mention of the model construction and external validation in the abstract and the results. I was directly given the conclusion that the new framework has good application in covid, TB and pneumothorax. So I would ask to revise the abstract as above within the limit.

→ Thank you for the suggestion. We agree with the reviewer that adding the contents of model development and results in the Abstract would be better for context. We revised the Abstract to add these contents within the word limits (< 150 words), as the reviewer suggested.

R2C2. (Results) Radiological assessment is not a definitive measure in terms of diagnosing TB, it seems the reference standard for TB in the external validation dataset is radiologist reading rather than bacteriological lab results. This is a limitation.

→ Thanks for the constructive comments. As commented, the current reference standard for diagnosis of tuberculosis is a bacteriological lab test, and performing experiments in a cohort consisting of radiologically diagnosed tuberculosis cases without laboratory confirmation can be a limitation. Therefore, we include new external validation data from another institution (Seoul National University Hospital [SNUH]) consisting of bacteriological lab test confirmed tuberculosis cases. In SNUH external data consisting of 727 normal and 535 bacteriological lab test confirmed tuberculosis cases, the results were remarkably consistent with our previous results, showing the stably improving as well as the superior performances to those with the fully supervised model, given the increasing unlabeled data, as shown in below Fig. R7.

Fig. R7. Tuberculosis diagnosis results: validation results in a cohort of bacteriological laboratory-confirmed tuberculosis cases. In a cohort from another core lab consisting of 727 normal cases and 535 confirmed tuberculosis cases, (a)(b) the model obtained with the *DISTL* framework offered performances superior to those of the fully supervised model trained with the same amount of labeled data. (c) Detailed diagnostic performances of the model trained with the proposed method in a cohort of bacteriological laboratory-confirmed tuberculosis cases. Shaded areas represent a 95% confidence interval (CI), and * denotes statistically significant ($p < 0.050$) superiority of the proposed framework. COVID-19, coronavirus disease 2019; AUC, area under the receiver operating characteristic curve; PPV, positive predictive value; NPV, negative predictive value.

We added these results in the new section “Validation in a cohort of bacteriological laboratory test confirmed tuberculosis cases.” in text and Fig. 9.

R2C3. The reference standards were mentioned for the external dataset of covid and TB on page 7, but no description of how pneumothorax was determined. I only found it in “Details of datasets for diagnosis” section. It would be good to include the reference standards all in the same place.

→ We appreciate the reviewer’s meticulous comment. Per your suggestion, we mentioned this on page 7 of the text.

R2C4. Figure-4, 5, and 8: Can you plot 95% CI as error bars?

→ Per your suggestion, we added 95% CI as shaded error bars to all graph plots, including Fig. 4, 5, 6, 8, 9, and Fig. S2.

R2C5. Code availability: the Github link does not work.

→ The link for code availability has been updated: <https://github.com/sangjoon-park/AI-Can-Self-Evolve>. This has been made a public repository.

References

1. Chen, T., Kornblith, S., Norouzi, M. & Hinton, G. A simple framework for contrastive learning of visual representations. in *International conference on machine learning* 1597-1607 (PMLR, 2020).
2. He, K., *et al.* Masked autoencoders are scalable vision learners. *arXiv preprint arXiv:2111.06377* (2021).
3. Caron, M., *et al.* Emerging properties in self-supervised vision transformers. in *Proceedings of the IEEE/CVF International Conference on Computer Vision* 9650-9660 (2021).
4. Li, X., *et al.* Learning to self-train for semi-supervised few-shot classification. *Advances in Neural Information Processing Systems* **32**(2019).
5. Yalniz, I.Z., Jégou, H., Chen, K., Paluri, M. & Mahajan, D. Billion-scale semi-supervised learning for image classification. *arXiv preprint arXiv:1905.00546* (2019).
6. Xie, Q., Luong, M.-T., Hovy, E. & Le, Q.V. Self-training with noisy student improves imagenet classification. in *Proceedings of the IEEE/CVF conference on computer vision and pattern recognition* 10687-10698 (2020).
7. He, J., Gu, J., Shen, J. & Ranzato, M.A. Revisiting self-training for neural sequence generation. *arXiv preprint arXiv:1909.13788* (2019).
8. Hinton, G., Vinyals, O. & Dean, J. Distilling the knowledge in a neural network. *arXiv preprint arXiv:1503.02531* **2**(2015).
9. Cubuk, E.D., Zoph, B., Shlens, J. & Le, Q.V. Randaugment: Practical data augmentation with no separate search. *arXiv preprint arXiv:1909.13719* **2**, 7 (2019).
10. Srivastava, N., Hinton, G., Krizhevsky, A., Sutskever, I. & Salakhutdinov, R. Dropout: a simple way to prevent neural networks from overfitting. *The journal of machine learning research* **15**, 1929-1958 (2014).
11. Huang, G., Sun, Y., Liu, Z., Sedra, D. & Weinberger, K.Q. Deep networks with stochastic depth. in *European conference on computer vision* 646-661 (Springer, 2016).
12. Irvin, J., *et al.* Chexpert: A large chest radiograph dataset with uncertainty labels and expert comparison. in *Proceedings of the AAAI conference on artificial intelligence*, Vol. 33 590-597 (2019).

Reviewers' Comments:

Reviewer #1:

Remarks to the Author:

I am happy with authors' response and the paper has been well revised that the contributions of the proposed approach are now well articulated. In this revision, the authors also added more experiment results to strengthen the impact of the study.

Reviewer #2:

Remarks to the Author:

The revision is comprehensive and responses fully addressed the comments from the first round of reviewers. I have no further comments.

NCOMMS-22-04319A

Response to the Reviewers

General Comments

We thank the editor and reviewers for the help with the revision. We have made the changes to the manuscript and supplementary information to meet the editorial request, which were documented in the version with the tracked changes. No changes to the major contents (e.g. introduction, main results, discussions, etc.) were made.

Reply to the Reviewer 1

I am happy with authors' response and the paper has been well revised that the contributions of the proposed approach are now well articulated. In this revision, the authors also added more experiment results to strengthen the impact of the study.

→ We appreciate the reviewer's understanding and the helpful comments in the previous revision. The reviewer's comments in the previous revision have significantly improved the clarity and impact of our paper.

Reply to the Reviewer 2

The revision is comprehensive and responses fully addressed the comments from the first round of reviewers. I have no further comments.

→ We are glad that our responses fully addressed the comments from the reviewers. Thank you for your helpful comments in the previous revision.